# Pulmonary Function and CT Scan Imaging at Low-Level Occupational Exposureto Asbestos

**DOI:** 10.3390/ijerph17010050

**Published:** 2019-12-19

**Authors:** Giannina Satta, Tiziana Serra, Federico Meloni, Achille Lazzarato, Alessandra Argiolas, Elisa Bosu, Antonella Coratza, Nicola Frau, Michele Lai, Luigi Isaia Lecca, Nicola Mascia, Ilaria Pilia, Veronica Piras, Giovanni Sferlazzo, Marcello Campagna, Pierluigi Cocco

**Affiliations:** Department of Medical Sciences and Public Health, Occupational Health unit, University of Cagliari, Asse didattico-Blocco I SS 554, 09042 Monserrato, Italy; tiserra3777@gmail.com (T.S.); federicomeloni@hotmail.it (F.M.); achillelazzarato@gmail.com (A.L.); ale.argiolas@hotmail.it (A.A.); eli.bosu@gmail.com (E.B.); coratzantonella@tiscali.it (A.C.); fraunicola91@gmail.com (N.F.); michele.lai21@gmail.com (M.L.); isaialecca@gmail.com (L.I.L.); mascia.nicola@hotmail.com (N.M.); cromatinap@gmail.com (I.P.); veronicapiras91@tiscali.it (V.P.); gio.sferlazzo@libero.it (G.S.); mam.campagna@gmail.com (M.C.); pcocco@unica.it (P.C.)

**Keywords:** asbestos, respiratory function, CT scan, retrospective exposure assessment

## Abstract

**Background**: In spite of the reduced exposure level, and its ban in numerous countries, compensation claims for asbestos-related diseases are far from decreasing. **Methods**: We used retrospective exposure assessment techniques to explore respiratory function and a computerized tomography (CT) scan in relation to past asbestos exposure in 115 male workers retired from an acrylic and polyester fiber plant. Based, on detailed information on exposure circumstances, we reconstructed a cumulative exposure estimate for each patient. **Results**: Time-weighted average exposure in our study population was 0.24 fibers/ml (95% confidence inteval (CI) 0.19–0.29), and the average cumulative exposure was 4.51 fibers/mL-years (95% CI 3.95–5.07). Exposure was elevated among maintenance workers, compared to other jobs (*p* = 0.00001). Respiratory function parameters did not vary in relation to the exposure estimates, nor to CT scan results. Risk of interstitial fibrosis showed a significant upward trend (Wald test for trend = 2.62, *p* = 0.009) with cumulative exposure to asbestos; risk associated with 5.26 fibers/mL-years or more, was 8-fold (95% CI 1.18–54.5). **Conclusions**: Our results suggest that a CT scan can detect pleuro-parenchymal lung alterations at asbestos exposure levels lower than previously thought, in absence of respiratory impairment. Further studies are required to validate our techniques of retrospective assessment of asbestos exposure.

## 1. Introduction

Asbestos-related disorders remain an occupational health concern in most developing countries, and continue to be a major cause of morbidity and mortality in developed countries, despite stricter regulations in the first, and the ban of asbestos use in the second during the last decades [1].

According to the 2012 International Agency for Research on Cancer IARC evaluation, asbestos is a well-established cause of mesothelioma and cancer of the lung, larynx, and ovary, as confirmed in several epidemiological studies in humans and experimental animals. Evidence of carcinogenicity is limited for pharynx, stomach and colon and rectum neoplasms [2]. Besides, exposure to asbestos fibers can induce a series of benign conditions, such as diffuse pleural thickening with or without calcifications, and interstitial fibrosis of the lung parenchyma (asbestosis) [1].

A program of surveillance of malignant and non-malignant asbestos-related diseases has been established in Italy for formerly exposed workers [3]. However, asbestos health effects are mainly related to the cumulative amount of inhaled fibers [4]; therefore, a reliable method of assessing past exposure at the individual level is a precondition of risk-differentiated surveillance several years later [5]. In fact, the several decades long, silent period of asbestos-related diseases, known as the latency period, entails a specific problem when planning such health surveillance programmes. The reconstruction of past exposures relies on data banks of industrial hygiene measurements to estimate airborne fiber concentration by occupation and industry [6], or for specific job tasks [7], or by job tasks in specific industries [8]; however, there have been no known attempts to validate such reconstruction in manufacturing and non-manufacturing settings. Besides, as epidemiological studies of asbestos related health effects have relied upon historical high-level measurements, there is large uncertainty on whether a threshold for such effects would exist and what would be [4]; also, comments have been made about the importance of computerized tomography (CT) scanning in identifying early damage to the lung parenchyma [9].

We elaborated retrospective estimates of lifetime exposure to asbestos in 115 workers with low-level exposure while employed in an acrylic and polyester fiber plant in Sardinia, Italy. Aim of this study is to investigate the correlation between such retrospective estimates, respiratory function tests, and high-resolution CT (HRCT) scan imaging.

## 2. Materials and Methods

### 2.1. Study Design and Participants

In recent years, following public concern, several surveillance programs of workers with past exposure to asbestos have been initiated at the regional level, with varying protocols, in most instances relying on a qualitative categorization of probability of asbestos exposure (possible, probable, certain).

Between February 2017 and July 2019, 115 workers, formerly exposed to asbestos in an acrylic and polyester fiber plant in Sardinia, Italy, attended our outpatient ambulatory services of Occupational Health at the University Hospital of Cagliari, Italy for a screening of their health status. The diagnostic protocol included a clinical exam, with special attention to the respiratory system, a short questionnaire on socio-demographics, tobacco smoking, alcohol drinking, and health history, and a detailed occupational history, with a careful description of the job tasks, their frequency and duration, a global spirometry test, and a diffusion lung capacity test with carbon monoxide (DLCO). All the patients also exhibited a recent (performed within the last three years) HRCT scan, which had been prescribed based upon the presence of respiratory symptoms, or clinical signs of pulmonary disease, and/or smoking status, and/or results of previous diagnostic imaging of the lungs. In most instances, the HRCT report was available, but not the related images, and, therefore, the application of the International Classification of Occupational and Environmental Respiratory Disorders (ICOERD) [10] was not feasible. Therefore, for the purposes of this analysis, as we used the grading proposed by Gamsu et al. [11], which we categorized as it follows: no pleuro-parenchymal alterations; Gamsu et al. grade I; Gamsu et al. grades II–VI. Three subjects for whom a recent CT scan was not available were excluded from this analysis.

All study subjects gave their informed consent for inclusion before enrolment in the study. The study was conducted in accordance with the Declaration of Helsinki, and the protocol was approved by the Ethics Committee of NP/2019/5553.

### 2.2. Retrospective Assessment of Occupational Exposure to Asbestos

Based on the detailed description of all job positions held by the study subjects in their work history, we identified job tasks implying direct manipulation, cut, or abrasive action on materials containing asbestos, as well as the type of material, whether friable or compact. For each such job task, we referred to the Datamyant database [7], and to available reports [8,12] to abstract the time-weighted (8-h) average concentration of asbestos fibers. For indirect exposure resulting from the contamination of the work environment due to maintenance activities, we based our calculations on the Ev@lutil database [6]. To calculate the individual time-weighted average exposure to asbestos (E*_i_*), we applied the airborne concentration (E*_n_*),corresponding to a specific task, to the fraction (T*_n_*) of working time (T_0_) dedicated to that specific task, and summed up the exposures from the individual tasks, according to the following algorithm [7,12]:E*_i_* = ∑E*_n_*× (T*_n_*/T_0_)(1)

We then calculated cumulative asbestos exposure (expressed as fiber-years [ff/mL-years]) by multiplying E*_i_* times the duration of exposure in years. Finally, we categorized the estimated exposure metrics into quartiles of the distribution among the whole study population.

### 2.3. Statistical Methods

For the purposes of analysis, we divided the study population into job categories, as plant operators and supervisors, service workers (including job in the laboratory, guards, stock and inventory clerks, raw material yard men, final product shipping clerks), and maintenance workers (mechanics, electricians, insulators). We conducted the analysis of variance (ANOVA) to compare the mean values of respiratory function parameters, and time-weighted average exposure to asbestos by job category, tobacco smoking (categorized into non-smokers, ex-smokers, and current smokers), and HRCT scan results. To reject the null hypothesis, we set the two-tailed α error threshold at *p* < 5%. We explored the relationship between metrics of exposure to asbestos, respiratory function and CT scan results by multiple regression analysis; covariates in the models included age, smoking, and estimated metrics of asbestos exposure. We used a case-control approach to explore the associations between job, duration of exposure to asbestos, and metrics of asbestos exposure with risk of interstitial fibrosis of the lung, based on the categorization of the HRCT scan as described above. We categorized time-weighted average exposure and cumulative exposure in quartiles; a further categorization of cumulative exposure was made by isolating from the top quartile subjects who accumulated 10 f/mL-years or more to identify a subgroup with a relatively higher exposure. We conducted logistic regression analysis with a CT scan Gamsu et al. grade I, and grades II–VI as alternative outcomes, to calculate the odds ratio (OR) and its 95% confidence interval (CI) associated with our asbestos exposure estimates, adjusting by age, and tobacco smoke. Tests for trend were calculated after continuous transformation of the categorical variables using the Wald statistics (*β/se_β_*), which follows the standard normal distribution. All the analyses were conducted using SPSS^®^version 16.0 (IBM Corporation, Armonk, NY, USA)

## 3. Results

Table 1 summarizes selected characteristics of the study participants. All participants were male; mean age was 65.5 years (*sd* 4.16), with maintenance workers slightly younger than either one of the other subgroups, and no difference between service workers and plant operators (analysis of variance: Fisher’s F = 4.02, *p* = 0.02). Exposure to asbestos among these workers lasted an average of 18.7 years (*sd* 5.37), and again it was slightly shorter in maintenance workers than either one of the other subgroups, and no difference between service workers and plant operators (analysis of variance: Fisher’s F = 4.44, *p* = 0.01). The majority of the study participants was represented by ex-smokers (63/115), 34 subjects reported never having smoked, and 18 were current smokers, with no substantial differences across subgroups (*χ*^2^ = 3.15, degrees of freedom = 4, *p* = 0.53).

As expected, over the whole study population, the time-weighted average exposure to asbestos was relatively low (mean = 0.24 f/mL, 95% confidence interval [CI] 0.19–0.30) and the average cumulative exposure was 4.51 f/mL-years (95% CI 3.40–5.61). Exposure was higher among maintenance workers (mean = 0.36 f/mL, 95% CI 0.30–0.43), compared to plant operators, and service workers (analysis of variance: Fisher’s F = 13.6, *p* = 0.00001), whose direct contact with asbestos was only occasional, and exposure had been mainly as bystanders or from the work environment.

Figure 1 shows the distribution of the time weighted average exposure to airborne asbestos among the whole study population.

Table 2. showsthe mean values of the respiratory functional parameters in all subjects and by job categories. The differences across job categories are random, suggesting lack of relevant effects of job category on the respiratory function.

Interstitial fibrosis with or without presence of pleural plaques (Gamsu et al. grades II–VI) was observed in 16 out of 112 workers (14.3%); the prevalence was higher among maintenance workers (10/59, 16.9%), than plant operators and service workers combined (6/53, 11.3%).

Respiratory function parameters did not vary in relation to the HRCT scan results (Table 3), nor were they related to the time weighted average or cumulative exposure to asbestos, after adjusting by age and smoking in a multiple regression analysis (Table 4).

As expected, vital capacity (VC), and forced expiratory volume in 1 second (FEV1) showed a linear decrease with age (*p* = 0.007, and *p* = 0.0009, respectively), and the residual volume (RV) showed a linear increase with age (*p* = 0.001). An effect of smoking as a categorical variable was also observed in reducing VC, FEV1, and DLCO, but it was much weaker. Pack-years in ever smokers, 63/81 of whom were ex-smokers, also did not show a clear effect; however, among 18 current smokers, FEV1 decreased with pack-years (*p* = 0.045); VC also decreased (*p* = 0.240), and RV increased (*p* = 0.104), though to a lesser extent, while DLCO was unaffected. At the low levels experienced by our study population, metrics of asbestos exposure did not affect pulmonary function. Apart from chance, we do not have an explanation for the observed RV decrease with duration of exposure (Table 4).

There was no increase in risk of lighter interstitial fibrosis of the lung (Gamsu et al. grade I) in relation to job as a maintenance worker, or to duration of employment in that job. Risk of Gamsu et al. grade II–VI interstitial fibrosis of the lung parenchyma was also not elevated among maintenance workers with reference to the other job subgroups combined; however, 20+ years employment as a maintenance worker was associated with a 2.7-fold increase in risk (OR = 2.7; 95% CI 0.77–9.61) (Table 5).

On the contrary, both the estimated time weighted average exposure, and cumulative exposure to airborne asbestos were significantly associated with increased odds of developing Gamsu et al. grades II–VI lung fibrosis, with significant upward trends in risk (time weighted average exposure: Wald test for trend = 2.84, *p* = 0.0045; cumulative exposure: Wald test for trend = 2.62, *p* = 0.009). Risk in the top quartile of cumulative exposure was increased 8-fold (OR = 8.0; 95% CI 1.18–54.5), and for cumulative exposures above 10 fibers/mL-years risk was increased 11-fold (OR=10.8; 95% CI 1.54–75.7) (Table 5).

## 4. Discussion

Occupational exposure to asbestos is of concern for millions of workers worldwide [13], and extending its ban would not manifest its effects for decades. Therefore, early identification of asbestos-related diseases is highly recommended, so that their evolution can still be reversed, arrested, or slowed [14]. According to the 2014 revision of the Helsinki criteria, CT scan, added to medical examination, structured questionnaires and checklists, and respiratory function tests, is the predominant screening test, especially among smokers and ex-smokers [14], as a more than additive interaction exists between the detrimental effects of asbestos and smoking [4,15,16,17,18,19]. In this paper, we included tobacco-smoking status as a covariate in multiple regression models to predict respiratory function parameters and risk of pulmonary fibrosis related to low-level asbestos exposure. We did not see much of an effect of smoking, which might be related to the effectiveness of anti-smoking campaigns in convincing smokers to quit their habit, particularly among workers previously exposed to asbestos. In fact, a large proportion of study subjects, well aware of the significant hazard posed by the interaction between the two risk factors, had quit smoking (63/115, 55%), while current smokers were much less represented (18/115, 16%).When we plotted respiratory function parameters vs. pack-years among current smokers, the expected decrease in FEV1 and increase in RV volumes became apparent.

Although asbestos related diseases are widely studied in occupational settings, considerable uncertainty exists on the dose-response curve at cumulative levels below 100 f/mL-years [20]. A threshold of 25 f/mL-years has been empirically suggested below which clinically relevant asbestos-related fibrosis would not show up [4]. However, the current difficulty in monitoring workers exposed to a broad range of airborne asbestos levels, so to allow full exploration of the dose response curve, limits the possibility of inference in this regard. Although criticized for the associated uncertainty [4], a retrospective exposure assessment methodology might be helpful in this regard, provided that a detailed description of tasks and average duration and frequency of the individual operations conveying asbestos exposure can be gathered from the individual and/or the company. Such information can link to existing industrial hygiene measurements in publicly available databases to come up with reasonable estimates of time-weighted average and cumulative exposure, as described above.

Using this approach, we explored the relationship between our exposure estimates, respiratory function parameters, and the HRCT scan results. Our results suggest that the odds of interstitial fibrosis of the lung and/or pleural plaques are elevated at estimated exposure levels well below 25 f/mL-years.

Risk estimates associated with parenchymal lung fibrosis were quite elevated, which might have resulted from the small number of cases, and particularly from having one case only whose exposure was in the reference quartile. The resulting imprecision in the risk estimates is, therefore, a weakness of our study.

A threshold exposure level for asbestos-related health effects can only be derived from using a definitely unexposed reference category. However, widely accepted safety principles do not allow performing X-ray procedures when unnecessary for diagnostic purposes. Therefore, as one purpose of our study was to assess whether our exposure estimates might have been related to CT scan images, we used the lowest quartile of the distribution of the exposure estimates over the whole study population.

If further large size surveys would confirm our findings, there might be two alternative explanations: either we underestimated exposure, which true levels might have been as much as 4 times higher to match the expected threshold for the effects we observed, or HRCT scan can detect early parenchymal changes at cumulative asbestos levels lower than the threshold thus far speculated. If the second explanation would apply, our results would also confirm the validity of retrospective exposure assessment in characterizing dose–response curves of asbestosis, lung cancer and mesothelioma [21].

A significant reduction in FVC and FEV1 volumes in subjects exposed to asbestos compared to unexposed subjects, and a significant FVC reduction in exposed 20+ years compared to subjects for a lesser duration have been previously reported [22]. In that study, the authors used duration as a surrogate for asbestos exposure, while, in our study, we did use retrospective exposure estimates and compared respiratory function tests by increasing category of time-weighted average and cumulative exposure. We could not detect changes in respiratory function related to the low exposure levels experienced by our study population nor to the HRCT scan results. Another study used a study design similar to ours to compare respiratory function test by cumulative exposure to asbestos, among subjects exposed to 49 f/mL-year on average, a level more than 10 times higher than that experienced by our study subjects. After excluding subjects with an abnormal CT scan, there was no correlation between cumulative exposure or duration of exposure, and lung function parameters and airways resistance [23].

The Helsinki consensus document recommends low-dose CT (LDCT) as the most suitable method for lung cancer screening among workers formerly exposed to asbestos [14], as it was proven effective in reducing mortality from lung cancer and all causes mortality among smokers [24]. Recently, programs offering free medical counseling and free health screening to asbestos workers have been initiated in several Italian regions, with different protocols, including chest radiographs interpreted according to the ILO criteria by specially trained B-readers; when high risk conditions apply, a CT scan is also prescribed [4]. The International Classification of Occupational and Environmental Respiratory Diseases (ICOERD) was proposed based on HRCT [10], which is considered to be most suitable to detect interstitial lesions, and it is routinely performed, while LDCT reading for pneumoconiotic lesions has not been standardized as yet [25]. Therefore, a recent (less than 3 years) HRCT scan was performed following medical prescription by all asbestos workers attending our outpatient ambulatory, which report we used for the purposes of this paper.

A major limitation in interpreting our findings is due to the self-selection of the study population, composed by asbestos workers who voluntarily attended out outpatient ambulatory. If attendance had been related to the HRCT result, selection bias might have artificially increased the prevalence of pulmonary fibrosis in respect to the whole population of asbestos workers. However, we applied our retrospective exposure assessment techniques in a standard way, blind to the HRCT result; therefore, although generalization of our findings is uncertain, the internal comparison between different exposure levels would maintain its validity.

As mentioned earlier, another limitation is due to the small size of our study population, which reduced the statistical power to detect marginal effects of low-level asbestos exposure, and smoking on lung function. The broad confidence interval of the risk estimates we calculated also point to low statistical power as an issue. Extending our approach to retrospective assessment of asbestos exposure to other similar occupational health outpatient services might be helpful in confirming or contrasting our findings.

## 5. Conclusions

We observed a strong association between our estimates of exposure to airborne asbestos and interstitial fibrosis of the lung at levels lower than what has been consideredpreviously. Further studies are warranted to validate the retrospective exposure assessment techniques we applied, before extending its application to health screening programs. Defining past exposure levels would allow selecting asbestos workers for whom surveillance should be stricter and more in depth for early detection of lung cancer, as well as for the right compensation of the physical and psychological damage caused by asbestos exposure.

## Figures and Tables

**Figure 1 ijerph-17-00050-f001:**
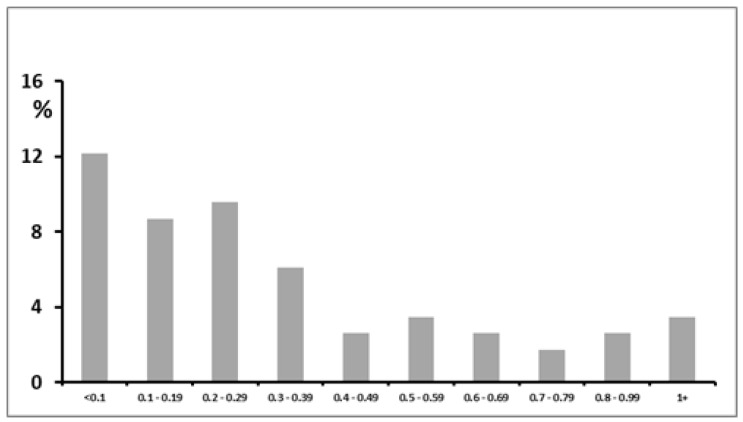
Frequency distribution of categories of time weighted average exposure to airborne asbestos among workers of an acrylic and polyester fiber plant.

**Table 1 ijerph-17-00050-t001:** Selected characteristicsof the study population (age, duration of exposure, tobacco smoking and time weighted average exposure to airborne asbestos) overall and by job category.

	Total Population (N = 115)	Plant Operators (N = 38)	Service Workers (N = 16)	Maintenance Workers (N = 61)	F (*p* Value)*χ^2^ (*p*Value)
Age (years, mean, *sd*)	65.5 (4.17)	66.4 (3.48)	67.1 (3.16)	64.6 (4.57)	4.02 (0.02)
Duration of exposure(years, mean, *sd*)	18.4 (5.37)	20.1 (4.96)	19.7 (2.63)	17.0 (5.82)	4.44 (0.02)
Ex-smokers + current smokers (n, rate)	61 (0.68)	28 (0.74)	12 (0.75)	41 (0.67)	3.15 (0.53)*
Time weighted average exposure to asbestos (fibers/mL; mean, *sd*)	0.24 (0.29)	0.13 (0.08)	0.07 (0.06)	0.36 (0.35)	13.6 (0.00001)

* significant *p*-value.

**Table 2 ijerph-17-00050-t002:** Mean values of respiratory functional parameters by job.

Job Categories	VC (L) Mean ± *sd*	FEV1 (L) Mean ± *sd*	RV (L) Mean ± *sd*	DLCO (mL/min/mmHg) Mean ± *sd*
Total population	3.76 ± 0.72	2.93 ± 0.58	2.35 ± 0.56	26.88 ± 6.52
Plant operators, supervisors	3.74 ± 0.64	2.88 ± 0.52	2.34 ± 0.67	27.86 ± 7.61
Service workers	3.40 ± 0.47	2.71 ± 0.50	2.43 ± 0.62	26.36 ± 5.22
maintenance workers	3.87 ± 0.80	3.02 ± 0.63	2.34 ± 0.47	26.41 ± 6.12
ANOVA: F (*p* value)	2.78 (0.07)	2.02 (0.137)	0.19 (0.829)	0.62 (0.539)

**Table 3 ijerph-17-00050-t003:** Respiratoryfunction parameters in relation to the grading of parenchymal fibrosis based on the result of the high-resolution computerized tomography (HRCT) scan.

HRCT	VC (L) Mean ± *sd*	FEV1 (L) Mean ± *sd*	RV (L) Mean ± *sd*	DLCO (mL/min/mmHg) Mean ± *sd*
Normal	3.80 ± 0.72	2.96 ± 0.54	2.32 ± 0.44	26.68 ± 5.98
Gamsu et al. grade I	3.77 ± 0.72	2.93 ± 0.61	2.45 ± 0.75	26.73 ± 5.59
Gamsu et al. grades II–VI	3.50 ± 0.65	2.71 ± 0.64	2.34 ± 0.34	28.59 ± 4.39
ANOVA: F (*p* value)	1.10 (0.336)	2.29 (0.308)	0.58 (0.560)	0.75 (0.473)

**Table 4 ijerph-17-00050-t004:** Coefficients from multiple regression models predicting respiratory function parameters in relation to age, smoking, and asbestos exposure metrics.

Covariates	VC (L) β, se	FEV1 (L) β, se	RV (L) β, se	DLCO (mL/min/mmHg) β, se
age	−0.043, 0.016*	−0.043, 0.013*	0.039, 0.012*	−0.073, 0.135
Smoking**	−0.042, 0.100	−0.028, 0.080	−0.077, 0.076	−1.226, 0.850
Time weighted average exposure to asbestos	0.066, 0.235	0.002, 0.186	0.320, 0.177	−0.317,1.972
Cumulative exposure to asbestos	−0.0008, 0.011	−0.002, 0.009	0.010, 0.008	0.004, 0.093
Duration of exposure to asbestos	−0.006, 0.013	0.001, 0.010	−0.022, 0.010*	0.001, 0.110

* *p*-value below 5%; ** smoking was a categorical variable (non-smokers, ex-smokers, current smokers) used as a continuous variable.

**Table 5 ijerph-17-00050-t005:** Risk of lung fibrosis associated with retrospective estimates of asbestos exposure.

Asbestos Exposure Metrics	Gamsu et al. Grade I			Gamsu et al. Grade II–VI		
	N	OR	95%CI	N	OR	95%CI
Maintenance workers	17/32	0.7	0.26–1.65	10/32	0.7	0.20–2.13
Maintenance workers ≤20 years	12/18	0.9	0.30–2.45	2/18	0.5	0.09–3.0
Maintenance workers ≥20 years	5/14	0.4	0.13–1.51	8/14	2.7	0.77–9.61
Time weighted average exposure						
≤0.06 f/mL	6/21	1.0	-	1/21	1.0	-
0.061–0.14 f/mL	10/18	2.4	0.69–8.59	3/18	4.7	0.41–53.9
0.15–0.29 f/mL	8/11	2.2	0.58–8.52	5/11	12.5	1.13–138
≥0.30 f/mL	11/11	4.0	1.06–15.1	7/11	18.4	1.75–193
Wald test for trend (p)		1.93	(0.054)		2.84	(0.0045)
Cumulative exposure						
≤1.09 f/mL-years	7/18	1.0	-	2/18	1.0	-
1.1–2.59 f/mL-years	9/19	1.2	0.36–4.37	1/19	0.5	0.04–6.75
2.6–5.25 f/mL-years	9/13	2.1	0.59–7.76	7/13	5.8	0.84–40.6
≥5.26 f/mL-years	10/11	2.5	0.68–9.01	7/11	8.0	1.18–54.5
Wald test for trend		1.39	(0.165)		2.62	(0.009)
≥10 f/mL-years	4/6	1.6	0.32–7.96	4/6	10.8	1.54–75.7

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
