# Peer review of "Pulmonary Function and CT Scan Imaging at Low-Level Occupational Exposureto Asbestos"

_ijerph, 2019, doi:10.3390/ijerph17010050_

Round 1

Reviewer 1 Report

Asbestos related diseases have a long latency period and lack a reliable method for early detection. There is also no way to determine who will develop malignant disease from exposure. In this study, Satta et. al. demonstrate that acrylic and polyester fiber factory workers exposed to slightly elevated time weighted average of approximately 0.24 fibers/ml air for the entire study group exhibited signs of lung fibrosis despite having cumulative asbestos exposure that were considered low. The interesting finding in this study was that there was no correlation between the level of exposure and interstitial lung fibrosis observed, although cumulative exposure greater than 20 years almost tripled the risk of developing higher grade fibrosis. Unfortunately, these exposure levels were retroactively inferred from employment history and  job description. Thus there is a likelihood that actual exposure levels may have been over or under estimated. This study emphasizes the need for continued monitoring of individuals exposed to asbestos no matter how low the exposure is considered. Despite the ban on asbestos usage in developed countries, developing countries are still using asbestos in restricted ways and therefore still stand the risk of having large numbers of people developing asbestos related diseases. 

The use of high resolution computed tomography provided valuable information to the comparison of the inferred asbestos exposure levels. The authors admit the limitations of their study, a small study size limits generalizations and extrapolations based on the results of the study. 

A control of factory workers not expected to be exposed to asbestos wold help determine what baseline asbestos exposure occurred for all workers at the plant.

There are several spacing issues after punctuations and in other places. e.g. line 32 space between 2012 and international; line 47 between industries and [8];

line 63: between 115 - workers, - formerly...etc.   

Author Response

A control of factory workers not expected to be exposed to asbestos would help determine what baseline asbestos exposure occurred for all workers at the plant.”

The reviewer is correct in the sense that only a reference group definitely unexposed would have allowed to determine with greater precision the asbestos threshold with detectable health effects. The baseline exposure level cannot be derived from comparison with an unexposed group, but from measurements or estimates such as the ones we used. On the other hand, safety precautions do not allow performing x-ray exams if not for diagnostic purposes or when there is clear indication (such as, for instance, for screening purposes among subjects with one or more risk factors including asbestos exposure). We added a short comment on this in the discussion section of the revised version (lines 218-223 on page 6 of the revised version).

“…several spacing issues after punctuations and in other places…”

In the revised version, we carefully reviewed the text and edited missing spaces, punctuation, and other digit mistakes.

Reviewer 2 Report

The paper describes the correlation between past asbestos exposure and current exploration of respiratory function and CT scan. Background and prior literature are properly presented.

Does the phrase “in most instances, only the HRCT report was available” means that in this patients no respiratory function tests were accessible or that there were no pictures? The authors have included in the study patients with a recent HRCT which has been performed due to respiratory symptoms but does the appearance of symptoms not contradict the concept of early diagnosis? This condition could cut a big informative piece of study sample. pg 6 line 194 The statement is not logical. The effectiveness of anti-smoking campaigns can affect the number of smokers or smoking duration, but cannot affect the harm caused by smoking or its impact on asbestosis.

Author Response

    1. “Does the phrase “in most instances, only the HRCT report was available” means that in this patients no respiratory function tests were accessible or that there were no pictures?

We apologize for the misunderstanding. The sentence the reviewer mentions was meant to clarify that in a number of cases we had no access to HRCT images. Respiratory function tests were always available. To make this point clearer, we modified the sentence (see lines 70-73 on page 2 of the revised version).

The authors have included in the study patients with a recent HRCT which has been performed due to respiratory symptoms but does the appearance of symptoms not contradict the concept of early diagnosis? This condition could cut a big informative piece of study sample.

As explained in the text (lines 68-70 on page 2; lines 231-237 on page 7), requests of a CT scan among workers formerly exposed to asbestos had to follow several preconditions, including being a heavy smoker or respiratory symptoms. We agree with the opinion of the reviewer that such strategy would cut an important piece of information, but again it was also important to comply with the existing guidelines prescribing that patients should be exposed to x-rays only for diagnostic purposes.

pg 6 line 194. The statement is not logical. The effectiveness of anti-smoking campaigns can affect the number of smokers or smoking duration, but cannot affect the harm caused by smoking or its impact on asbestosis”.

We are grateful to the reviewer for having detected the illogical statement. In the revised version (line 191-194, on page 6) we amended the sentence by making clear that the effect of the anti-smoking campaigns was related to a high fraction of smokers ex-exposed to asbestos quitting their habit. 

Reviewer 3 Report

Correlation between pulmonary functional parameters and asbestos exposure merit investigation. Recently, Lopatin et al. showed that long duration at lowlevel asbestos exposure is associated with further declines in pulmonary function[1]. Here, authors analyzed respiratory function and CT scan in relation to past asbestos exposure and drew related conclusions. Several problems need to be addressed;

1.    Details need to be clarified;

In 2016, Schikowsky C, et al. found no significant association between lung function and asbestos exposure. Here, authors carried about similar research. Would authors please explain discrepancies between two manuscripts, regarding to experimental design and results part?

2.    Table 1: please analyze correlation between parameters and cohorts.

3.    Fig.1 Please analyze difference between groups.

Reference

1.    Lopatin S, Tsay JC, AddrizzoHarris D, Munger JS, Pass H, Rom WN. Reduced lung function in smokers in a lung cancer screening cohort with asbestos exposure and pleural plaques. Am J Ind Med. 2016; 59:178–185.

2.    Schikowsky C, et al. Lung function not affected by asbestos exposure in workers with normal Computed Tomography scan. Am J Ind Med. 2017 May;60(5):422-431.

Author Response

Lopatin et al. showed that long duration at lowlevel asbestos exposure is associated with further declines in pulmonary function”.

Lopatin et al., 2016 observed a significant reduction in FVC and FEV1 volumes in subjects exposed to asbestos compared to unexposed subjects, and a significant FVC reduction in exposed 20+ years compared to subjects for a lesser duration. Their study did not include any assessment of the exposure level. On the other hand, in our study we did use retrospective exposure estimates and compared respiratory function tests by increasing category of time-weighted average and cumulative exposure. As we reported in the text, respiratory function tests were not affected at the exposure level experienced by our study population. In the revised version, we cited this paper in a short sentence added to the discussion (lines 230-236 on page 7).

In 2016, Schikowsky C, et al. found no significant association between lung function and asbestos exposure. Here, authors carried about similar research. Would authors please explain discrepancies between two manuscripts, regarding to experimental design and results part?

We are grateful to the reviewer for having brought to our attention these two papers. Schikowsky et al. studied a cohort of workers with a high level of cumulative exposure to asbestos (mean cumulative asbestos exposure was 49 f/ml-year), and explored the correlation between lung function tests and metrics of asbestos exposure, in subjects with a normal CT scan. Their results did not show an association between asbestos exposure and lung function impairment, when there were no parenchymal or pleural changes. In our study subjects, who experienced a cumulative exposure to asbestos more than 10 times lower, we explored the correlation over the whole study population, and not limited to those with normal CT scan only. We also did not find a correlation. In the revised version, we added a short sentence to present the average cumulative exposure in our study population (which was missing in the previous version) for sake of comparison with the Schikowsky et al. study (lines 129-130 on page 3). Besides, we added a new paragraph comparing the experimental design and the results of our study with that of Schikowsky et al. (lines 237-242 on page 7).

Table 1: please analyze correlation between parameters and cohorts.”

We did compare the value of the parameters by study subgroup, and reported the results in the text for the variables in Table 1 (see lines 116-132 on page 3-4), and those for the respiratory function tests in Tables 2 and 3. In the revised version, we now show the statistics for each parameter also in Table 1. We hope this matches the reviewer suggestion.

Fig.1: Please analyze difference between groups.”

Figure 1 was meant to show graphically the distribution of time weighted average exposure to asbestos among the whole study population. There was no specific hypothesis to test, and therefore no statistical model to apply. We did analyse the difference in mean time weighted average exposure by study subgroups, and we used the analysis of variance to test the differences. The results of such comparison are described in the text (lines 128-133 on page 3-4), and they are reported in Table 1.

Round 2

Reviewer 2 Report

Accept as it

Reviewer 3 Report

Authors improved the quality of manuscript.